# The Anti-Listeria Activity of *Pseudomonas fluorescens* Isolated from the Horticultural Environment in New Zealand

**DOI:** 10.3390/pathogens12020349

**Published:** 2023-02-19

**Authors:** Vathsala Mohan, Reginald Wibisono, Saili Chalke, Graham Fletcher, Françoise Leroi

**Affiliations:** 1The New Zealand Institute for Plant and Food Research, Private Bag, Auckland 92169, New Zealand; 2IFREMER, MASAE Microbiologie Aliment Santé Environnement, F-44000 Nantes, France

**Keywords:** *Pseudomonas fluorecens*, *Listeria* spp., biocontrol agents, *tss*-T6SS secretion system, anti-listeria activity, structural proteome analysis

## Abstract

Beneficial bacteria with antibacterial properties are attractive alternatives to chemical-based antibacterial or bactericidal agents. Our study sourced such bacteria from horticultural produce and environments to explore the mechanisms of their antimicrobial properties. Five strains of *Pseudomonas fluorescens* were studied that possessed antibacterial activity against the pathogen *Listeria monocytogenes*. The vegetative culture of these strains (*Pseudomonas fluorescens*-PFR46I06, *Pseudomonas fluorescens*-PFR46H06, *Pseudomonas fluorescens*-PFR46H07, *Pseudomonas fluorescens*-PFR46H08 and *Pseudomonas fluorescens*-PFR46H09) were tested against *Listeria monocytogenes* (n = 31), *Listeria seeligeri* (n = 1) and *Listeria innocua* (n = 1) isolated from seafood and horticultural sources and from clinical cases (n = 2) using solid media coculture and liquid media coculture. All *Listeria* strains were inhibited by all strains of *P. fluorescens*; however, *P. fluorescens*-PFR46H07, *P. fluorescens*-PFR46H08 and *P. fluorescens*-PFR46H09 on solid media showed good inhibition, with average zones of inhibition of 14.8 mm, 15.1 mm and 18.2 mm, respectively, and the other two strains and *P. fluorescens*-PFR46H09 had a significantly greater zone of inhibition than the others (*p* < 0.05). There was no inhibition observed in liquid media coculture or in *P. fluorescens* culture supernatants against *Listeria* spp. by any of the *P. fluorescens* strains. Therefore, we hypothesized that the structural apparatus that causes cell-to-cell contact may play a role in the ejection of ant-listeria molecules on solid media to inhibit *Listeria* isolates, and we investigated the structural protein differences using whole-cell lysate proteomics. We paid special attention to the type VI secretion system (*TSS*-T6SS) for the transfer of effector proteins or bacteriocins. We found significant differences in the peptide profiles and protein summaries between these isolates’ lysates, and PFR46H06 and PFR46H07 possessed the fewest secretion system structural proteins (12 and 11, respectively), while PFR46H08 and PFR46H09 had 18 each. *P. fluorescens*-PFR46H09, which showed the highest antimicrobial effect, had nine *tss*-T6SS structural proteins compared to only four in the other three strains.

## 1. Introduction

There is a strong demand for sustainable alternative technologies to improve the quality and safety of fresh produce and food production. As part of established appropriate horticultural practices for the production and sale of safe food products, growers rely on physical washing and chemical sanitizers. The use of chemical-based sanitizers can pose health risks and cause undesirable environmental effects. This has resulted in increasing concern over current control measures, and chemicals such as chlorine are being banned in some countries [1]. Alternative control measures for pathogens are therefore being sought, including other synthetic chemicals such as organophosphates, carbamates and pyrethroids [2]; however, unfortunately, controversies have been raised about the use of these chemicals, and the other mode of pathogen control is to implement biocontrol measures. Biological control agents (BCAs) are living organisms that are used to control unwanted organisms. They have been used in various fields of biology, particularly in entomology and plant pathology against pests and microbial pathogens to suppress their populations [3]. One such unwanted pathogen on fresh produce including fruits and vegetables is *Listeria monocytogenes* (*L. monocytogenes*). Biological control agents are an attractive alternative to implement a sustainable natural control measure against *L. monocytogenes* on fresh produce.

*L. monocytogenes* is a Gram-positive, non-sporulating bacillus that is ubiquitous in nature and has been isolated from a wide range of sources, including horticultural produce, processed foods, dairy products, silage and soils [4,5]. As a facultative intracellular pathogen, *L. monocytogenes* can cause invasive diseases, such as meningoencephalitis, sepsis and gastroenteritis, in immunocompromised humans, as well as miscarriage in pregnant women. It causes disease in several farm animals, including cows, sheep, pigs and goats [6,7]. Human disease occurs as a result of direct contact with infected animals or due to ingestion of contaminated food products [8,9,10].

Listeriosis outbreaks from fresh produce have been reported widely globally. For example, in 2010, the Texas Department of State Health Services (DSHS) reported a listeriosis outbreak that affected patients aged 56 to 93 years (n = 10) in which five patients died within three months of infection, as reviewed by Zhu, Gooneratne [11]. Similarly, in the US, consumption of contaminated lemons caused a listeriosis outbreak in 28 states that resulted in 33 deaths and 147 hospitalisations. The PFGE typing of the isolates was reported to match *L. monocytogenes* isolates from cantaloupe, as reviewed by Zhu, Gooneratne [11]. Another well-known outbreak is the caramel apple outbreak that occurred in the US in 2014, with a cost of 35 lives, and in 2016, there was another outbreak from packaged salads in Ohio [11]. Several fresh-produce-associated listeriosis outbreaks have been reported in fresh vegetables and processed fresh produce products, as reviewed by Macarisin, Sheth [12], which necessitates a tailored sustainable control measure to be put in place to control human listeriosis.

Biological control agents (BCAs) refer to the use of non-pathogenic microorganisms and/or their metabolites to extend the shelf life of food and to improve its microbiological quality [13,14]. In particular, lactic acid bacteria (LAB) are used to aid in food preservation. LABs have a long history of use in fermented foods, making them attractive choices for use in bio-preservation, particularly for protection against *L. monocytogenes* [15]. Similarly, other groups of organisms are used to control pathogens, for example, *Pseudomonas fluorescens* is a plant-growth-promoting rhizobacterium (PGPR) and has been identified as a potential BCA for bacterial diseases in plants [3,16]. Its saprophytic nature and natural soil adaptation abilities permit robust survival in soil. Certain strains have proven to be potent BCAs that suppress plant diseases by protecting the seeds and roots from fungal infection [17].

*P. fluorescens* comprises a group of saprophytes that commonly colonize soil, water and plant surface environments. *P. fluorescens* is a Gram-negative, rod-shaped bacterium that secretes a soluble fluorescent pigment called fluorescein, particularly under conditions of low iron availability [18]. Most *P. fluorescens* strains are obligate aerobes, although some strains take up NO_3_ for respiration in place of O_2_. This species is motile, with multiple polar flagella. *P. fluorescens* has been demonstrated to grow well in mineral salt media with carbon sources [19]. *P. fluorescens* produces secondary metabolites, including antibiotics, siderophores and hydrogen cyanide [20]. Rapid colonization through competitive exclusion of pathogens is the main mechanism of action by which *P. fluorescens* inhibits pathogens in the rhizosphere (reviewed by Haas and Defago [21]). *P. fluorescens* is not generally considered a bacterial pathogen in humans, as reviewed by Scales, Dickson [22], which makes bacteria a safer option to act as a biocontrol agent. Type VI secretion system (*tss*-T6SS) components have been shown to be the main mechanistic apparatus of the interaction with plants and potentially other competitors through in multiple genomic, proteomic and transcriptomic studies on *P. fluorescens* strains [23,24,25,26]. In addition, the type VI secretion system (*tss*-T6SS) in Gram-negative *Proteobacteriaceae* is an important molecular mechanistic apparatus crucial for microbial interactions with a virulence role that exhibits selective advantages in response to danger signals [6,27,28,29]. Furthermore, the T6SS was revealed to have structural similarities with the tail and puncturing device of the bacteriophage T4, which provide selective advantages to annihilate competitors [30,31] and deliver a great variety of effectors with a broad range of activities [32,33,34]. This also provides an advantage of using *P. fluorescens* as a BCA because as a Gram-negative bacteria, it can penetrate Gram-positive organisms and inhibit them [35].

Considering the harmful effects of chemical sanitizers and the potential of BCAs in controlling foodborne pathogens, we aimed to isolate a resident bacteria from fresh apple produce with bio-preservative properties and to study their potential as BCA candidates.

## 2. Materials and Methods

### 2.1. Screening for Protective Bacteria in Horticultural Produce

We divided the study into two parts. The first part involved isolation of resident bacterial species that have antibacterial activity (potential protective bacteria) from horticultural produce and/or their processing environments. The second part involved strain characterization: (1) species identification by 16SrRNA sequencing, (2) structural proteome analysis using liquid chromatography with tandem mass spectrometry (LC-MS/MS) and (3) comparison of the secretion of structural proteins with special reference to the *tss*-T6SS secretion system among the strains that exhibited antibacterial activity.

This study was conducted in an effort to isolate resident bacteria with potential biocontrol properties from an apple packhouse in New Zealand. The method for setting up a screening study was adapted from previous studies conducted by Kieser and Wassmer [36] and Whitehead, Julious [37]. The authors applied an 80% upper confidence interval and showed that and overall sample size of between 20 and 40 corresponds to a standardized size effect for 90% power based on a standard sample calculation. Our study included a total of 34 samples (apples and swabs). We conducted sampling in an apple processing packhouse where three apples were collected from each wash cycle; three sample bags were collected from dirty apples and clean apples (n = 3/wash cycle/bag; 3 sample bags /cycle; a total of 27 apples), and swabs from the critical control points were identified during processing (n = 7) in Gisborne, New Zealand. The apples (n = 3 per bag of samples) were hand-massaged in 400 mL of Butterfield broth (Difco-BD Becton Dickinson, and Company, Sparks, MD 21152, USA), and the swabs were stomached for 2 min in 100 mL Butterfield broth (recommended for food, dairy and environmental sample processing for microbial revival and/or growth by the FDA-approved Bacteriological Analytical Manual (Bacteriological Analytical Manual (BAM) | FDA). The processed samples were incubated at 30 °C for 24 h, and 1 mL aliquots from the incubated broth samples were used to screen for resident bacteria with biopreservative properties.

### 2.2. Screening for Resident Biocontrol Bacteria Using Listeria Pour Plates

*L. monocytogenes* Scott A was grown in tryptic soy broth with 0.6% yeast extract (TSBYE) (Difco-BD, Becton, Dickinson and Company, USA) for 24 h at 37 °C, and 1 mL of the 24 h culture was used to make the *L. monocytogenes* pour plates using TSAYE (tryptic soy agar with yeast extract) (Difco-BD, Becton, Dickinson and Company, USA). After solidification, 100 µL from the sample broths was spread using plate spreaders and left uncovered in a level II biosafety cabinet for 2 h to allow the samples to be absorbed. The spread pour plates were incubated at three different temperatures (20 °C, 30 °C and 37 °C) in an effort to provide good coverage to capture the resident bacteria with potential biocontrol properties that grow at these temperatures.

### 2.3. Selection of Presumptive Biocontrol Colonies

Bacterial colonies that produced an identifiable zone of inhibition with Scott A were picked from the pour plates using sterile one-microliter loops (Mediwire, UK) and regrown in TSBYE broth at the same respective temperatures as above (20 °C, 30 °C and 37 °C). The cultures from the broth were then streaked onto TSAYE plates after 24 h of incubation. If a mixture of cultures was found, single colonies of each morphology were streaked on *Listeria* CHROMagar^TM^ (CHROMagar, 75006 Paris, France) to discriminate the *L. monocytogenes* colonies from non-*Listeria* cultures, as the spread plates had both *L. monocytogenes* and the test biocontrol (BCA) organisms. The non-*Listeria* cultures were picked from the CHROMagar^TM^ and grown in TSBYE broth for 24 h at 20 °C, 30 °C and 37 °C, and 100 µL of each culture was spread onto *L. monocytogenes* Scott A pour plates to confirm the inhibitory activity of the respective colony cultures. The colonies that showed a zone of inhibition were further purified following the same procedure until a pure colony with a zone of inhibition was obtained. The pure colonies were propagated in brain–heart infusion agar (BHI, Difco-BD Becton, Dickinson and Company, USA) plates and stored at −85 °C in 50% glycerol broth until further testing.

### 2.4. Zone of Inhibition Using Agar Gel Diffusion Test

The purified presumptive or test BCA cultures (n = 14) that grew at 30 °C (no biocontrol properties were observed at other temperatures) and that showed some degree of inhibition were subjected to an agar gel diffusion test to confirm their antibacterial effect against *L. monocytogenes* Scott A and against other strains of *L. monocytogenes* from the Plant & Food Research culture collection (PFR18C07, PFR18D01 and PFR18D05). Three different disc diffusion coculture methods were tested: (1) pour plates with test BCA supernatants, (2) pour plates with test BCA vegetative cultures and (3) mixed cultures of test BCA and *L. monocytogenes* on *Listeria* CHROMagar. The *L. monocytogenes* strains were washed and adjusted to an optical density (OD) of 0.5 at 600 nm, and 1 mL of the culture was used in pour plates of TSAYE agar. Once solid, 4 mm diameter holes were aseptically punched into the agar with a sterile borer, and the agar plugs were removed using sterile pipette tips. The presumptive BCA cultures were grown in TSBYE broth for 24 h at 30 °C and centrifuged at 4 °C (Eppendorf, 5810R, Eppendorf AG, Barkhousenweg1, Hamburg, Germany) at 3220× *g* for 10 min. The supernatants were separated, filter-sterilized using 0.2 micron syringe filters (Sartorius, Thermo Fisher Scientific, Auckland, New Zealand) and stored at 4 °C before further testing. The cell pellets were washed twice and resuspended with 0.1% tryptone (Difco-BD, Becton, Dickinson and Company, USA) and adjusted to an OD value of 0.5 at 600 nm, and 30 µL of each isolate’s supernatant and vegetative cultures was added to holes of the pour plates. A chloramphenicol disc (30 µg, Mast Diagnostics, Mast group Ltd., Merseyside, United Kingdom) was placed in the center of each plate as a positive-susceptibility control. Plates were incubated at 30 °C and 20 °C, and zones of inhibition were recorded over a period of 5 days (0, 1, 2, 3, 4 and 5 days). The radius from the edge of the well to the edge of the clear zone was measured using a Vernier caliper (ROK Precision Instrument, Shenzhen, China). In the case of irregular edges, radii on all four sides of the inhibition zone were measured, and an average was calculated. Alternatively, vegetative cells (10 µL) were placed on top of the pour plates (without holes) and incubated to measure the zone of inhibition. Similarly, the pellets of *L. monocytogenes* strains and test BCA cultures (30 µL each) were mixed and plated (10 µL of the mixture) on to the CHROMagar plates to observe the inhibition apart from pour plates.

### 2.5. Culture Supernatant Susceptibility and Liquid Coculture Tests

BCA control cultures proven to be listeriolytic (*Carnobacterium maltaromaticum* 1944, *C. maltaromaticum* 2003, *C. maltaromaticum*, *Carnobacterium divergens* 2122, *Leuconostoc gelidum* and *Lactococcus piscium*) were used as positive control supernatants, in addition to antibiotic discs, for supernatant susceptibility testing (the supernatants were prepared following the same technique described above to test BCA cultures). The cultures were supplied by Institut Français de Recherche pour l’Exploitation de la Mer (IFREMER), Laboratoire de Génie Alimentaire, Nantes, France. Thirty µL of the sterile supernatant from all cultures and the proven BCA cultures was dispensed into the wells (replicates of two plates were tested), left in the biosafety cabinet for 1 h for the liquid to be fully absorbed and then incubated at 20 °C and 30 °C for 5 days (days 0, 1, 2, 3, 4 and 5); the zone of inhibition was recorded on all 5 days.

### 2.6. Species Identification Using 16SrRNA Sequence Analysis

All pure BCA test cultures (n = 5) with a recognizable zone of inhibition were grown in TSAYE plates for 24 h, and DNA was extracted using 2% Chelex solution (Chelex 100 resin, BioRad laboratory, Hercules, CA, USA). The colonies (2–3 mm) were picked using sterile loops, suspended in 2% Chelex solution and heat-treated at 100 °C for 10 min. The treated solution was cooled to room temperature, then centrifuged (Eppendorf, 5424R) at 21,130× *g* for 5 min, and 250 µL of the supernatant was separated and stored for PCR reactions. Isolates with DNA A_260_/A_280_ ratios between 1.8 and 2.0 were taken for PCR amplification, while DNA isolation was repeated for those not fulfilling this quality criterion until it was achieved. The DNA samples were amplified using universal 16S rRNA bacterial primers 27F (5′-AGAGTTTGATCCTGGCTCAG-3′) and 1541R (5′-AAGGAGGTGATCCAGCCGCA-3′) [38,39].

The PCR reaction mix comprised 10 μL of 1x PCR buffer (Thermo Fisher Scientific), 2 µL of 10 μM DNTP mix (Thermo Fisher Scientific), 1 µL of 10 μM of each primer (forward and reverse, IDT, Australia), 1 µL platinum Taq polymerase (Thermo Fisher Scientific) (one unit per reaction), MgCl_2_ 1.5 mM (Thermo Fisher Scientific) and 10 ng/µL of DNA. The reaction mix was prepared to a final volume of 50 µL with sterile Milli Q water. The PCR reaction was carried out in an Eppendorf Master Gradient Cycler with the following conditions: initial denaturation at 95 °C for 2 min, 95 °C for 30 s, annealing at 55 °C for 30 s, elongation at 72 °C for 2 min and a further elongation of 10 min. The reaction was carried out for 35 cycles.

PCR products were viewed under 1% agarose gel (Ultrapure™ Agarose, Invitrogen, Thermo Fisher Scientific, MA, USA) stained with RedSafe nucleic acid staining solution (JH Science/iNtRON Biotechnology USA) using a gel documentation system (BioRad, Hercules, CA, USA) to confirm the presence the PCR product (approximately 1464 bp). PCR products were purified using a QIAquick PCR purification kit (Qiagen), and the products were quantified using a Nanodrop and sent to Macrogen Inc., Korea, at a concentration of 20 µg/µL for sequencing. The FASTA sequences of all the cultures were blasted using NCBI BLAST blasting suite (https://blast.ncbi.nlm.nih.gov/Blast.cgi, last accessed on 7 January 2023). The sequences were identified to the genus level, where unambiguous high identity and coverage of ≥95 to 98% were taken as the criteria for determining the genus of the sequenced DNA [40]. Sequence analyses of the isolates were carried out in Geneious v10.1 software, and a neighbor-joining phylogenetic tree was constructed in MEGA v7 software using the bootstrap method (1000 replications). The forward and reverse sequences were compared individually, and a consensus was used for comparison and phylogenetic tree construction.

### 2.7. Fluorescence Activity and Zone of Inhibition against Listeria spp. of the Sequenced Test BCA Cultures

Isolates that were 16SrRNA-sequenced and identified as *Pseudomonas fluorescens* were subsequently tested for fluorescence activity (grown on BHI agar plates) by UV irradiation (Molecular Imager^®^, Gel Doc™ XR+ Imaging system, model: Universal Hood II, BioRad laboratories Inc., USA) and image capture. The plates that were cultured for 24 h were kept inside the UV chamber and viewed for fluorescence. Subsequently, these isolates were investigated for their inhibitory effects using randomly selected well characterized *Listeria monocytogenes* isolates (Table 1; n = 35; using the methods described elsewhere in this manuscript) sourced from seafood (n = 17), horticultural produce (n = 14) and clinical isolates (n = 2), plus one *L. innocua* and one *L. seeligeri*. A positive control chloramphenicol disc (30 µg) was used in the center of the plate for each strain. *L. innocua* and *L. seeligeri* were used in the trial to determine whether the organisms are capable of inhibiting other *Listeria* spp.

### 2.8. Testing for Antibacterial Activity of P. fluorescens in Liquid Media Coculture

Cultures of *L. monocytogenes* strains were washed and adjusted to ODs of 0.5 at 600 nm, as were *P. fluorescens* isolates. Both microbes were cocultured in TSBYE broth by adding 50 µL of each strain of *P. fluorescens* individually to each *Listeria* strain and serially diluted to 10^−7^ with a replicate of 2 using 96-well plates (Nunc U-bottom 96-well plates, Agilent Technologies, New Zealand). The coculture was incubated at 30 °C, 20 °C and 37 °C and observed from day 0 of coculture for up to 15 days. After each time point of incubation, 10 μL of the coculture from all dilutions was plated onto CHROMagar^TM^ *Listeria* and TSAYE plates to observe the growth of blue colonies for *Listeria* spp. on CHROMagar^TM^ *Listeria* and white or creamy colonies on TSAYE for *P. fluorescens.*

### 2.9. Proteome Analysis of P. fluorescens

The isolates identified as *P. fluorescens* using 16S rRNA (n = 4; 3 isolates with a zone of inhibition larger than 10 mm and one with a zone of inhibition smaller than 10 mm) were subjected to proteomic analysis using whole-cell lysates. Proteome analysis was used to confirm the species, in addition to 16S rRNA sequencing using their core proteins blasted against the proteome database. As the isolate PFR46I06 did not exhibit fluorescence and did not have a profound zone of inhibition, only PFR46H06, PFR46H07, PFR46H08 and PFR46H09 were subjected to proteome analysis. The cultures were grown in BHI broth for 24 h and centrifuged at 3200× *g* for 20 min at 4 °C (Eppendorf, 5424R). The cell pellets were transported on ice for proteome analysis at the Mass Spectrometry Centre, Faculty of Sciences, University of Auckland, using a nanoLC-equipped TripleTOF 6600 mass spectrometer (ABSCIEX, USA) using the information-dependent acquisition (IDA) method. The sample process involved cysteine alkylation using iodoacetamide, and the samples were digested with trypsin with urea denaturation. The ProteinPilot data were searched using the UniProt protein database of *Pseudomonas fluorescens* sequences (October 2018). The summaries of proteins, peptides and distinct peptides with modifications were analyzed with special reference to the secretion system, *tss*-T6SS. The mass spectrometry proteomics data were deposited with the ProteomeXchange Consortium via the PRIDE [1] partner repository with the dataset identifier PXD019965.

### 2.10. Statistical Analysis

Statistical analyses were carried out using R version 3.5.1. Each experiment was carried out twice (2 replicates per isolate) for agar gel diffusion tests and for the liquid coculture methods. The mean, standard deviation and standard errors were calculated, and one-way and two-way ANOVAs and post hoc Fisher’s least significant difference (LSD), as well as Bonferroni with an alpha error of 0.05, were calculated using the Agricolae^TM^ package in R for each strain of *P. fluorescens* against 35 strains of *Listeria* spp. A logistic regression model was used to compare the *L. monocytogenes* strains and the zone of inhibition produced by each strain of *P. fluorescens*. This model consisted of three *Pseudomonas* strains (PFR46H07, PFR46H08 and PFR46H09) that produced zones of inhibition of greater than 10 mm on all tested *Listeria* strains. The model included the *Listeria* strains and *Pseudomonas* strains as influencing variables. (*Listeria* strains vs. PFR46H07+ PFR46H08+ PFR46H09).

## 3. Results and Discussion

*L. monocytogenes* is a Gram-positive zoonotic pathogen that is found in a wide variety of sources, including fresh vegetable produce/horticultural produce, processed foods, dairy products, silage and soils [4,5]. Owing to the ability of *L. monocytogenes* to cause disease in vulnerable populations [6,7], its presence in food products and its ability to gain resistance to numerous chemical sanitizers [1,2,41], an alternative strategy has become a necessity. In the present study, we aimed to isolate a resident bacterial species with listeriolytic activity from fresh horticultural produce to be used as a potential biocontrol organism against *L. monocytogenes* and investigate its potential as a BCA to control Listeria species in horticultural environments and processing plants.

We isolated 5 strains of *P. fluorescens*, of which 3 showed significant anti-listeria activity in solid media, inhibiting all 35 tested strains of *Listeria* spp. The zones of inhibition were more pronounced at 20 °C than 30 °C, at which temperature the zones were hazy and not very clear. The PFR46H09 strain was significantly (*p* = 0.02) more inhibitory than the other strains.

### 3.1. BCA Bacterial Culture Screening, 16SrRNA Identification and Fluorescence Testing

Of the 27 apples and 7 environmental swab samples screened for resident bacteria with biocontrol characteristics, five cultures (PFR46H06, PFR46H07, PFR46H08, PFR46H09 and PFR46I06) were isolated with listericidal activity in the initial screening test. Of the five isolates, three (PFR46H07, PFR46H08 and PFR46H09) exhibited recognizable zones of inhibition (larger than 10 mm), while the other two showed zones of inhibition smaller than 5 mm. The five cultures were subjected to 16S rRNA gene sequencing for species identification, and *P. fluorescens* was identified based on the sequence identity score (98%). Figure 1 shows the neighbor-Joining phylogenetic tree of the five isolates, along with best-matching blast sequences that had 98% matching sequence identity. Four of these five cultures, when grown on BHI agar plates, had a light-green color and fluoresced under UV light, while PFR46I06 did not show profoundly bright fluorescence.

### 3.2. Zone of Inhibition

Five isolates (PFR46I06, PFR46H06, PFR46H07, PFR46H08 and PFR46H09) showed a detectable zone of inhibition at both 20 and 30 °C; however, the zones of inhibition were clearer and more defined on the culture plates at 20 °C after 48 h than 24 h. Isolates PFR46H06 and PFR46I06 showed very small zones of inhibition towards some strains (less than 5 mm) and were not inhibitory against the majority of the *L. monocytogenes* isolates. Therefore, the three strains that had a minimum zone of inhibition of 10 mm and one isolate with a zone of inhibition smaller than 10 mm were selected for further experiments. The average size of inhibition zones for the culture plates grown at 20 and 30 °C were 14.8 mm, 15.1 mm and 18.2 mm against all *Listeria* strains for PFR46H07, PFR46H08 and PFR46H09, respectively. One-way ANOVA comparing individual strains against 35 *L. monocytogenes* strains showed no significant difference in the zone of inhibition between the three strains. Figure 2 shows the inhibition zones produced by the three strains against 35 *Listeria* strains. One-way ANOVA individually comparing the zone of inhibition variances between the three *P. fluorescens* strains showed no significant differences against *Listeria* spp. However, in the logistic regression model with Listeria, the intercept was significant (*p* < 0.00), and the inhibition zones produced towards the *L. monocytogenes* strains PFR18C07 and PFR18D05 differed significantly from those produced against the other *Listeria* strains. The model compared *P. fluorescens* and *Listeria* spp., taking one pair as a comparison pair, with PFR5A10 taken as a comparison strain against all three strains of *P. fluorescens* (Table 2). In contrast, the supernatants did not show any inhibition against any *Listeria* strain.

### 3.3. Anti-Listeria Activity in Liquid Media in Coculture

There was little or no evidence of inhibition of *Listeria* spp. when cocultured with any of the *P. fluorescens* isolates in liquid media from day 1 or after 24 h. The plated cultures had an equal number of both strains according to macroscopic observation after incubation. The cultures were very slimy and ropy and became difficult to pipette or plate as the number of days in liquid coculture increased. In contrast, in the 0-day plated culture, inhibition on solid media increased as the number of days of incubation increased, which was evidenced by creamy, irregular colonies observed following 20 and 30 °C incubation over small to medium round *Listeria* colonies that were not blue and did not grow well when streaked on CHROMagar and produced inhibition on *Listeria* pour plates.

### 3.4. Proteome Analysis of P. fluorescens Strains

We analyzed four inhibitory isolates: PFR46H06, PFR46H07, PFR46H08 and PFR46H09. Figure 3 shows the false discovery rates of proteins at 1%, 5% and 10% error rates compared with global protein databases. The proteome analysis further confirmed the isolates as *P. fluorescens*, providing further support for the 16SrRNA sequencing results. All four isolates showed different protein hits, with 1781 in PFR46H06, 2030 in PFR46H07, 2228 in PFR46H08 and 1994 in PFR46H09. The lowest number of proteins (1781) was in PFR46H06, followed by PFR46H09, the strain with the strongest antimicrobial properties, suggesting that its proteomes may be small compared with those of the other isolates. Due to the liquid coculture results, our main interest was in investigating the secretion systems with special reference to the *tss*-T6SS in each of the isolates to investigate the structural protein composition.

Generally, *Pseudomonas* is a noted psychrotrophic genus of spoilage organisms found in soil, water and vegetation [19,42,43]. Pseudomonads are also commonly found in unpasteurized milk and dairy products [42]. Although pseudomonads are reported to enhance the growth of non-pathogenic and pathogenic bacteria in dairy products [42,43,44], studies also suggest that *P. fluorescens* has antibacterial activity against certain foodborne pathogens such as *L. monocytogenes* [17,45,46]. It should be noted that *P. fluorescens* has also been proposed as a BCA [3], and fluorescent pseudomonads have been studied for biocontrol research since 1970 using a process known as bacterization [47].

In our study, we observed remarkable inhibition of Listeria species by *P. fluoresecens*, particularly in solid media in contrast to liquid culture or liquid media, which was intriguing. Similarly, Farrag and Marth [46] reported *P. fluorescens* to only moderately inhibit *L. monocytogenes* in skim milk stored at 7 and 13 °C; they also reported an enhancement of growth of *L. monocytogenes* Scott A in the presence of *P. fluorescens* P26 after 7 days of incubation at 7 °C. A similar observation was made by Douglas and Schimdt [48] at 10 °C; however, in this trial, after 14 days, the populations declined compared with controls. Both studies suggested a limited inhibitory effect of *P. fluorescens* against *L. monocytogenes* in liquid media at low temperatures. In contrast, another study evaluated the listeriolytic activity of Pseudomonas sp. using the agar spot method with PGY agar plates, reporting significant inhibition at 20 °C, which is in agreement with the results of our current study [49]. In the present study, we have shown that there are differences among *P. fluorescens* strains, as only three of five strains caused reasonable inhibition, and different *L. monocytogenes* strains had different responses towards these strains in solid media. Other Gram-negative bacteria such as *E. coli* are known to inhibit other bacterial species via bacteriocins, namely colicins, which are secreted into the medium and are lethal to other bacterial cells (as reviewed by Cascales, Buchanan [50]); these proteins are transported through nutrient transporters located on the outer membrane, as well as a group of inner-membrane and periplasmic proteins [51]. However, in our study and in other previous studies, liquid coculture was not very successful in inhibiting Listeria; therefore, we speculate that *P. fluorescens* may require a physical structure for the transport of bacteriocins or effector proteins, unlike colicins. Our speculations are also based on research studies on competitive exclusion of pathogens, which has been identified as the main mechanism of action of *P. fluorescens* to inhibit enemies, as reviewed by Haas and Defago [21]. It should also be noted that genomic, proteomic and transcriptomic studies of *P. fluorescens* identified the type VI secretion system (*tss*-T6SS) components as the major contact apparatus for the interaction with plants and other bacteria [6,23,24,25,26,27]. The *tss*-T6SS has been shown to exhibit selective advantage in response to danger signals [28,29] as a puncturing device such as bacteriophage T4 to annihilate competitors [30,31] to deliver effectors [32,33,34]. Based on these studies and the importance of *tss*-T6SS structural components, as well as the absence of inhibition in liquid media in our study, we paid special attention to the structural proteins of *P. fluorescens* isolates in our study, which shed some light on the structural proteins of the tss-T6SS component.

We compared different secretion systems, including fimbria- and flagella-related proteins, phage and phage-related proteins and hemolysin proteins, and found substantial differences in the number of proteins among the four isolates. Appendix A lists all the proteins that were detected in the proteomes of each *P. fluorescens* isolate, while Table 3 lists the secretion system proteins. PFR46H06 and PFR46H07 possessed the fewest secretion proteins (12 and 11, respectively), while PFR46H08 and PFR46H09 each had 18. PFR46H09, which showed the greatest antimicrobial effect, had nine *tss*-T6SS proteins compared to just four in the other three strains (Table 4).

Predicted *tss*-T6SS protein ImpK was present in all four isolates (Table 3 and Table 4). However, there were notable modifications in the protein. In PFR46H06 and PFR46H09, a protein modification substituting the amino acid from R to N was found at the 95th position, while in PFR46H07, this R-to-N substitution occurred at the 92nd position. In PFR46H08, the predicted ImpK protein had a number of substitutions and modifications (Appendix A for individual strains with distinct peptide summaries are submitted): at 63, A to G; at 64, N to M; at 67, V to M; at 68, E to D; at 70, V to M; and at 95, R to N.

The next protein that we observed closely was the *rhs* gene protein of the *tss*-T6SS. Protein modifications were found at positions 37 (T to V) and 427 (S to T) in PFR46H06, PFR46H07 and PFR46H08, while in PFR46H09, there was an additional modification found at the 427th position (S to T). Similarly, protein modifications were found in other secretion proteins in the *tss*-T6SS system in PFR46H07, PFR46H08 and PFR46H09, such as a type VI secretion protein similar to the type VI secretion system contractile sheath small subunit *Vip*A in other *Pseudomonas* spp. [32,52]. In general, the number of secretion proteins found in PFR46H08 and PFR46H09 was greater than that in the other two isolates. The *tss*-T6SS system proteins ClpV2, TssK1, TssK, VipB and ImpM were present in PFR46H09, while the other isolates lacked one or more of these proteins (Table 3).

Other researchers conducted proteomic studies with different research aims than that of the present study. For example, Paul, Dineshkumar [53] examined the proteome of *P. fluorescens* MSP-393 in an effort to investigate the osmotolerance and/or saline stress levels of this strain for use in agricultural production. Similarly, Kim, Silby [54] examined the proteome of *P. fluorescens* strain Pf01 and identified the non-annotated protein coding genes. In contrast, we carried out whole-cell lysate proteome analysis and looked at the structural proteins of secretion systems, fimbria- and flagella-related proteins and the *tss*-T6SS secretion system to understand the differences between these strains, with special reference to the *tss*-T6SS system, as it is a recently identified secretion system [52,55].

In general, Gram-negative bacteria have been shown to utilize various secretion systems to deliver molecules to other bacterial and/or target cells, as well as extracellular surfaces; these systems are considered important virulence factors, as reviewed by Costa, Felisberto-Rodrigues [56]. Given that *L. monocytogenes* is a Gram-positive bacterium, it is intriguing to observe that *P. fluorescens* inhibits *L. monocytogenes*; we speculate that the *tss*-T6SS, which is responsive to danger signals [28,29], could be the major player. Because a physical contact is necessary to deliver the effectors to destroy other competing species, we believe that *tss*-T6SS plays a critical role in listeriolytic activity and transports a wide variety of effectors; without such a transport apparatus, this inhibition may have not been possible (reviewed by Filloux [57]).

In our attempt to investigate the major structural proteins that are crucial for the mechanistic apparatus, we found that the *tss*-T6SS system needs approximately 15 conserved and closely linked genes to form a functional apparatus [58] and that this apparatus is required to transport the hemolysin-coregulated protein and the valine–glycine repeat (Vgr) family proteins [59]. Recent X-ray crystallographic studies [60,61] suggested that these proteins are similar to bacteriophage tube and tail-spike proteins, with researchers speculating that *tss*-T6SS could be evolutionarily, structurally and mechanistically related to bacteriophage, which is bactericidal. The fundamental understanding of the mode of action of the *tss*-T6SS changed significantly after the discovery that numerous critical components of *tss*-T6SS are functionally homologous to the structural components of contractile phage tails [56]. The Hcp-predicted protein group was shown to be a structural homolog of phage tube proteins. In *Pseudomonas aeruginosa*, Hcp1 was shown to be the most abundant *tss*-T6SS secreted protein and has been structurally shown to be a donut-shaped hexamer [59]. These hexamers were shown to stack on top of each other head-to-tail to form continuous tubes in crystals that are identical to the external and internal proteins of the bacteriophage T4 tail tube [62]. As found in *P. aeruginosa*, in our proteomic study, we observed several phage-related tube, tail and sheath proteins present in all four strains; however, each strain was different in terms of its respective protein summary, which suggest that the genes for the tss-T6SS mechanistic apparatus were expressed and that the apparatus was fully formed in the *P fluorescens* isolates that inhibited Listeria species. The strains of *P. fluorescens* had proteins that are either T7 tail tube proteins or homologs of T7 tail-, tube-and sheath-associated proteins. It should also be noted that the *tss*-T6SS system has been recognized as the sixth major protein secretion system that is post-transcriptionally activated by cell-damage-derived signals via the RetS/Gac/Rsm pathway (reviewed in [63]).

While observing the whole structural proteome, we observed that PFR46H09 possessed fewer total proteins but exhibited relatively larger inhibition zones than other strains. This strain also possessed relatively more *tss*-T6SS proteins and flagella- and phage-related structural proteins, which could have been the rationale behind its enhanced inhibition. PFR46H09 colonies were irregular in shape, and the presence of numerous flagella-related proteins explains the movement on the solid agar media compared with other strains. Another essential conserved *tss*-T6SS protein, TssE, has been shown to be homologous to the T4 phage baseplate [60,64,65]. In our study, we identified the following baseplate proteins: TssL in PFR46H07 and Tssk and TssK1 proteins in PFR46H09 (Table 3). Bonemann, Pietrosiuk [66] showed that VipA (TssB) and VipB proteins form a tubular polymer, and Leiman, Basler [60] showed that the overall structure resembled the T4 phage polysheath and can be disassembled by ClpV substrate protein. In our study, PFR46H09 revealed ClpV2, and all four strains showed VipB proteins (Appendix A).

We searched the literature for other Gram-negative bacteria that may use the *tss*-T6SS system. A similar mechanism has been explained in Vibrio species; for example, the *tss*-T6SS in *Vibrio cholerae* [67] was shown to resemble a long phage tail that is attached to a cell envelope through an anchor. The tail was shown to have two conformations: one extended and one contracted—which is similar to the VipA/VipB sheath [66,67]. The contracted sheath structures were, in general, shorter and wider, and the *tss*-T6SS sheath assembly in *V. cholerae* was shown to take about 20–30 s before the sheath contracted to about half its length in less than 5 ms. This sheath was shown to disassemble in the presence of ClpV [67]. The *tss*-T6SS dynamics were studied in a detailed manner using live cell imaging in *V. cholerae*, *P. aeruginosa* and *Escherichia coli* [67,68,69]. However, the roles of the *tss*-T6SS conserved proteins are still deemed to be largely unknown [55], although *tss*-T6SS has been shown to have a different mode of action from other secretion systems [67,68,69].

*Likewise, the tss*-T6SS-predicted protein ImpK was found to be present in all four of our isolates, and this protein is thought to be similar to *E. coli* outer-membrane protein and to the flagellar torque-generating protein that was discovered in a study published in 2003 [70]. There were several notable differences in the protein modifications in all four isolates. In PFR46H06 and PFR46H09, a protein modification from R to N was found at the 95th position, while in PFR46H07, the substitution occurred at the 92nd position, and in PFR46H08, a number of substitutions and modifications were detected compared with the global *P. fluorescens* protein databases (Appendix A). However, the functional alterations have to be investigated to study the impact of the modifications.

Another predicted protein family is the rhs protein family, which is known to be widespread in Gram-negative bacteria [71,72,73]; all of our strains possessed the *rhs* gene of the *tss*-T6SS, with modifications in some strains compared with the reference strains in the UniprotKB database. PFR46H09 possessed a protein modification at the 427th position from S to T, which made this protein different from the other three strains. Similarly, protein modifications were found in the *tss*-T6SS contractile sheath small subunit *Vip*A, which was also identified in previous studies [32,52]. As previously mentioned, functional analysis is necessary to study the impact of these modifications.

Other predicted proteins include the Imp family proteins, which were present in all four isolates. A 2003 study reported a putative operon of 14 genes in *Rhizobium leguminosarum* strain RBL5523 and named them impA–impN [70]. The predicted ImpK (present in all strains) protein resembles an outer-membrane protein gene of *E. coli*; the authors suggested that the Imp system encoded components of a secretion apparatus and that the proteins dependent on Imp genes blocked colonization/infection processes in pea plants [70], which emphasizes the importance of the mechanistic contact for inhibition, as well as enhancement.

PFR46H06, which minimally inhibited *Listeria*, lacked two secretion proteins that were present in the three other inhibitory strains (type II secretion system protein F and type VI secretion protein). This may explain its poor inhibition or lack thereof. Among the *tss*-T6SS proteins, PFR46H06 possessed a transposon-mediated virulence protein, vasK, which was not found in the other isolates, and this protein has been proposed be associated with the *tss*-T6SS system, which is required for the cytotoxicity of *V. cholera* cells toward *Dictyostelium* amoebae [74].

Although in this study, we evaluated secretion system structural proteins, some of the limitations of our study include the use of Butterfield solution/broth, TSB broth and agar media, which are general-purpose nutritional media that are non-selective and non-differential for bacterial growth. We acknowledge that if any fastidious listeriolytic bacteria were to be present in the samples, they would not have been grown and/or would have required specific nutrition to be isolated in the culture media. However, in a way, it is advantageous as it reduces tedious culture processing required to isolate fastidious organisms that may possess listeriolytic activity.

We also acknowledge that these strains were cultured in TSBYE without aiming for conditional expressions of the genes. Therefore, these strains may exhibit different characteristics, at least in terms of inhibition of other bacteria under different environmental conditions, and their effector proteomes may differ depending on the conditions. We observed little or no inhibition of *Listeria* in liquid media, as also observed by Silverman with *P. aeruginosa* poorly expressing the *tss*-T6SS system in liquid medium, while it was expressed during surface growth [75]. This warrants more research on the tss-T6SS structural proteins, their modifications and the impact on functionality.

## 4. Conclusions and Future Directions

In the present study, we evaluated the listeriolytic activity of five strains of *P. fluorescens* isolated from horticultural environments through zone of inhibition testing, identifying three strains with strong inhibition and two strains that produced a minimal listeriolytic effect. We investigated the secretion system structural proteins, and we speculate that some probable mechanisms in the antibacterial effects of the *P. fluorescens* isolates may be mediated through the *tss*-T6SS system in each strain. However, the machinery of the *tss*-T6SS system in *P. fluorescens* needs further investigation in order to better understand the exact mechanisms of action and their potential benefits for use as an efficient antibacterial and/or listeriolytic agent, as well as the listeriolytic protein that is produced and transported, which can be applied to food processing environments to investigate their listeriolystic activity under natural environmental conditions. Furthermore, the genes that are involved in the formation of structural components may be subjected to mutagenesis experimentation, which can shed more light on the mechanistic aspects of the *tss*-TS66 system. PFR46H09 is an ideal candidate to conduct knockout gene experimentation, as this strain exhibited the most significant listeriolytic properties among the three studied strains.

## Figures and Tables

**Figure 1 pathogens-12-00349-f001:**
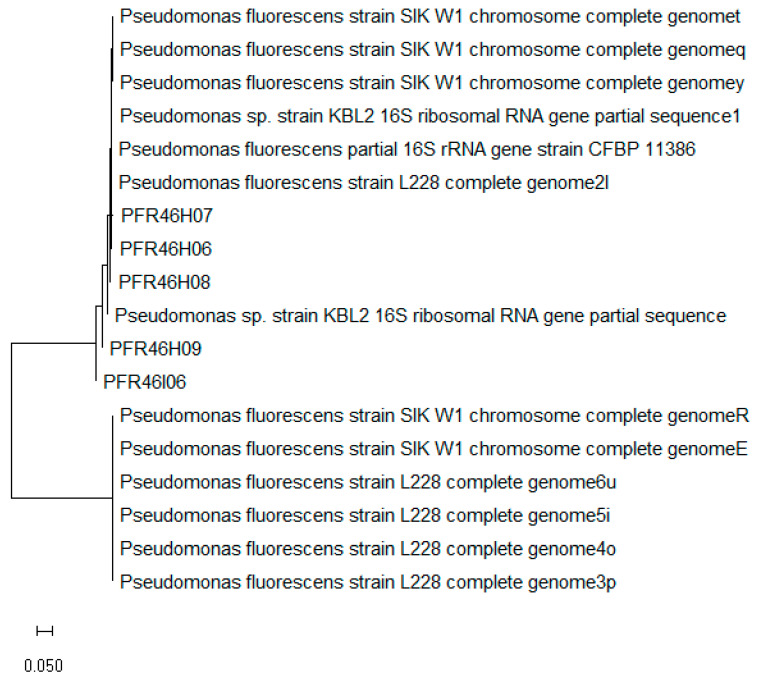
Neighbor-joining tree of the 16SrRNA sequences of 13 reference *Pseudomonas fluorescens* isolates that had a sequence identity above 95% in blast analysis, along with the five isolates of *P. fluorescens* isolated in this study constructed using MEGA v7 with a bootstrap value of 1000 replications. The forward and reverse sequences were compared individually, and a consensus was used for the comparison and phylogenetic tree construction.

**Figure 2 pathogens-12-00349-f002:**
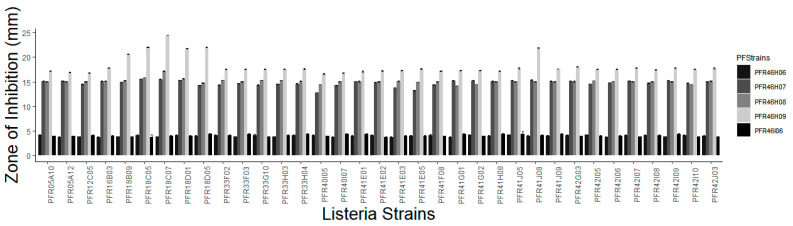
Graph of inhibition zones (average of two replicates) produced by five *Pseudomonas fluorescens* strains against 31 *Listeria monocytogenes* strains, one *Listeria innocua* strain (PFR05A10) and one *Listeria seeligeri* strain (PFR05A12) collected from seafood and horticultural sources in New Zealand and two international clinical isolates (*L. monocytogenes* Scott A = PFE12C05 and ATCC 49594 = PFE16B03). PFStrains = *Pseudomonas fluorescens* strains.

**Figure 3 pathogens-12-00349-f003:**
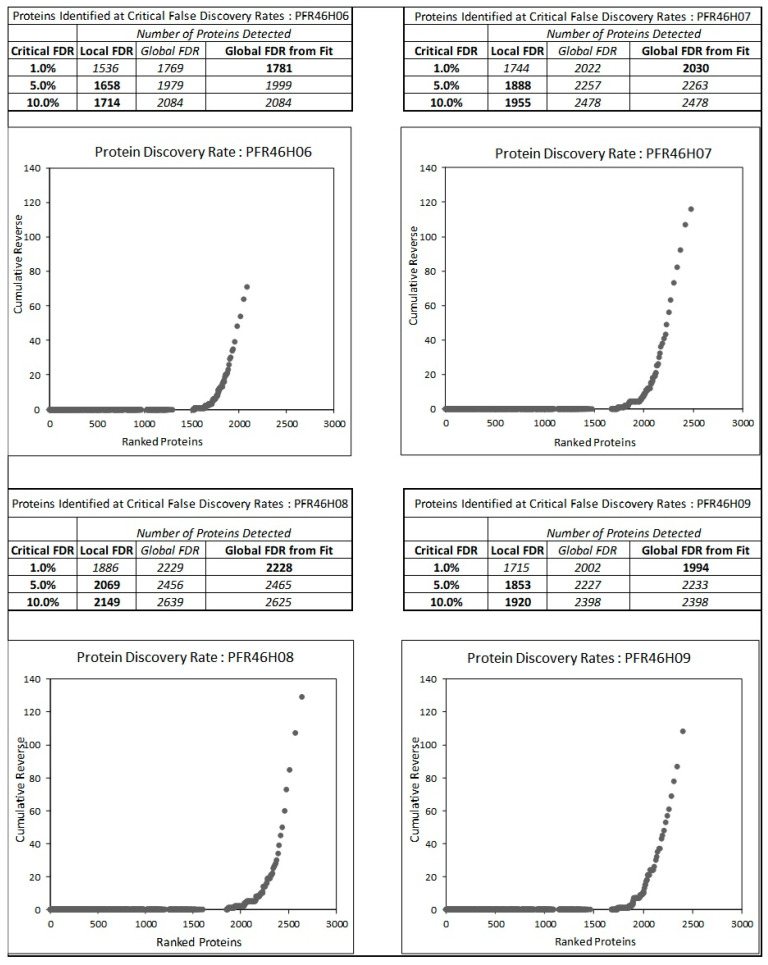
The number of proteins at false discovery rates (FDR) of 1, 5 and 10% for the *Pseudomonas fluorescens* isolates. The total number of proteins varied in each isolate, with PFR46H06, PFR46H07, PFR46H08 and PFR46H09 showing 1781, 2030, 2228 and 1994 proteins, respectively.

**Table 1 pathogens-12-00349-t001:** *Listeria monocytogenes*, *L. seeligeri* and *L. innocua* strains used in the study and their New Zealand sources.

No.	Strain	Organism	Source
1	PFR05A12	*L. seeligeri*	Vegetable
2	PFR12C05	*L. monocytogenes* Scott A	Clinical isolate
3	PFR16B03	*L. monocytogenes* ATCC strain 49594	Clinical isolate
4	PFR18B09	*L. monocytogenes*	Seafood processing environment
5	PFR18C05	*L. monocytogenes*	Seafood processing environment
6	PFR18C07	*L. monocytogenes*	Seafood processing environment
7	PFR18D01	*L. monocytogenes*	Seafood processing environment
8	PFR18D05	*L. monocytogenes*	Seafood processing environment
9	PFR33F02	*L. monocytogenes*	Seafood processing environment
10	PFR33F03	*L. monocytogenes*	Seafood processing environment
11	PFR33H03	*L. monocytogenes*	Seafood processing environment
12	PFR33H04	*L. monocytogenes*	Seafood processing environment
13	PFR33I04	*L. monocytogenes*	Seafood processing environment
14	PFR40I05	*L. monocytogenes*	Horticultural source
15	PFR40I07	*L. monocytogenes*	Horticultural environment
16	PFR41E01	*L. monocytogenes*	Horticultural environment
17	PFR41E02	*L. monocytogenes*	Horticultural environment
18	PFR41E03	*L. monocytogenes*	Horticultural environment
19	PFR41E05	*L. monocytogenes*	Horticultural environment
20	PFR41F08	*L. monocytogenes*	Horticultural environment
21	PFR41G01	*L. monocytogenes*	Horticultural environment
22	PFR41G02	*L. monocytogenes*	Horticultural environment
23	PFR41H07	*L. monocytogenes*	Horticultural environment
24	PFR41J05	*L. monocytogenes*	Horticultural environment
25	PFR41J08	*L. monocytogenes*	Horticultural environment
26	PFR41J09	*L. monocytogenes*	Horticultural environment
27	PFR42G03	*L. monocytogenes*	Horticultural environment
28	PFR42I05	*L. monocytogenes*	Seafood processing environment
29	PFR42I06	*L. monocytogenes*	Seafood processing environment
30	PFR42I07	*L. monocytogenes*	Seafood processing environment
31	PFR42I08	*L. monocytogenes*	Seafood processing environment
32	PFR42I09	*L. monocytogenes*	Seafood processing environment
33	PFR42I10	*L. monocytogenes*	Seafood processing environment
34	PFR42J03	*L. monocytogenes*	Seafood processing environment
35	PFR05A10	*L. innocua*	Processed vegetable

**Table 2 pathogens-12-00349-t002:** Logistic regression model of the inhibition zones produced by three *Pseudomonas fluorescens* strains against 33 *Listeria monocytogenes* strains, one *Listeria innocua* strain (PFR05A10) and one *Listeria seeligeri* strain (PFR05A12) collected from New Zealand seafood, seafood processing environments and horticultural sources (PFR05A10 was considered as a reference in the model by default). Bold fonts represent *p* values that are either highly significant (underlined, *p* < 0.05, ***), significant (underlined, *p* < 0.05, *) or borderline (*p* ≥ 0.05 and *p* ≤ 0.10).

*Listeria* Strains					
	**Estimate**	**Std. Error**	**z Value**	**Pr(>|z|)**	
PFR05A10 (Intercept)	2.32	0.58	4.00	** 0.00 **	***
PFR05A12	0.13	0.78	0.17	0.86	
PFR12C05	−0.18	0.73	−0.25	0.80	
PFR16B03	−0.34	0.70	−0.48	0.63	
PFR18B09	−1.06	0.64	−1.66	**0.10**	**.**
PFR18C05	−1.13	0.63	−1.80	**0.07**	**.**
** PFR18C07 **	−1.48	0.61	−2.43	** 0.02 **	*****
PFR18D01	−1.16	0.63	−1.85	**0.06**	**.**
** PFR18D05 **	−1.41	0.62	−2.28	** 0.02 **	*****
PFR33F02	−0.49	0.69	−0.71	0.48	
PFR33F03	−0.37	0.70	−0.52	0.60	
PFR33G10	−0.47	0.69	−0.68	0.50	
PFR33H03	−0.43	0.70	−0.62	0.54	
PFR33H04	−0.44	0.70	−0.64	0.53	
PFR40I05	−0.83	0.67	−1.23	0.22	
PFR40I07	−0.28	0.72	−0.40	0.69	
PFR41E01	0.05	0.76	0.07	0.95	
PFR41E02	−0.13	0.73	−0.18	0.86	
PFR41E03	−0.62	0.68	−0.91	0.37	
PFR41E05	−0.91	0.66	−1.38	0.17	
PFR41F08	−0.35	0.71	−0.49	0.62	
PFR41G01	0.01	0.75	0.01	0.99	
PFR41G02	0.01	0.75	0.01	0.99	
PFR41H08	−0.08	0.74	−0.11	0.91	
PFR41J05	−0.17	0.72	−0.24	0.81	
PFR41J08	−1.17	0.63	−1.87	**0.06**	**.**
PFR41J09	−0.23	0.72	−0.31	0.75	
PFR42G03	−0.36	0.70	−0.51	0.61	
PFR42I05	−0.43	0.70	−0.61	0.54	
PFR42I06	−0.32	0.71	−0.46	0.65	
PFR42I07	−0.26	0.71	−0.37	0.72	
PFR42I08	−0.31	0.71	−0.43	0.67	
PFR42I09	−0.26	0.71	−0.36	0.72	
PFR42I10	−0.33	0.71	−0.46	0.65	
PFR42J03	−0.29	0.71	−0.41	0.68	

**Table 3 pathogens-12-00349-t003:** Summary of the different secretion system proteins detected in four *Pseudomonas fluorescens* proteomes: PFR46H06, PFR46H07, PFR46H08 and PFR46H09.

Secretion Protein	PFR46H06	PFR46H07	PFR46H08	PFR46H09
Type I restriction enzyme R protein	-	-	-	-
Type I restriction enzyme R protein	+ REVERSED	-	+	-
Type I restriction–modification protein subunit M REVERSED	+	-	-	-
Type I secretion membrane fusion protein	-	-	+	+
Type I secretion outer-membrane protein	-	+	-	+
Type I secretion outer-membrane protein tolC	+	-	-	-
Type I secretion system ATP-binding protein PrsD	-	-	+	-
Type I secretion system membrane fusion protein PrsE	-	-	+	+
Type I secretion system permease/ATPase	-	-	+	+
Type II secretion pseudopilin HxcU REVERSED	-	-	+	+
Type II secretion system protein F	-	+	+ REVERSED	+ REVERSED
Type II secretion system protein GspJ REVERSED	-	-	-	+
Type III effector	-	-	+	+
Type III pantothenate kinase, coaX	+	-	-	-
Type III PLP-dependent enzyme	+	+	+	-
Type III restriction system endonuclease REVERSED	+	+	+	+
Type III secretion system transcriptional regulator RspS REVERSED	-	-	+	-
Type IV pilus response regulator PilH	+	+	+	+
Type IV secretion protein Rhs	+	+	+	+
Type IVB pilus formation outer-membrane protein, R64 PilN family	-	+	+	-
Type VI polysaccharide biosynthesis protein VipB/TviC	+	+	+	+
Type VI secretion ATPase, ClpV2	-	-	-	+
Type VI secretion protein	-	+	+	+ REVERSED
Type VI secretion protein TssK1 REVERSED	-	-	-	+
Type VI secretion protein TssL	-	+	-	-
Type VI secretion protein VasK REVERSED	+	-	-	-
Type VI secretion system baseplate subunit TssK REVERSED	-	-	-	+
Type VI secretion system protein ImpK	+	+	+	+
Type VI secretion system protein ImpM	+	-	+	+

**Table 4 pathogens-12-00349-t004:** Summary of total proteins, secretion system proteins and important substitutions in secretion system proteins detected in PFR46H06, PFR46H07, PFR46H08 and PFR46H09 *Pseudomonas fluorescens* strains.

Protein	PFR46H06	PFR46H07	PFR46H08	PFR46H09
Total number of proteins	1781	2030	2228	1994
Secretion system proteins	12	11	18	18
Number of *tss*-T6SS proteins	4	4	4	9
Phage-related tube, tail and sheath proteins	Present	Present	Present	Present
Type I restriction enzyme proteins	2	1	5	4
Type II secretion proteins	Absent	1	2	3
Type III secretion/effector proteins	3	2	4	2
Type IV pilus response regulator PilH	Present	Present	Present	Present
Type IV secretion protein Rhs	Substitutions in positions 37 (T to V) and 427 (S to T)	Substitutions in positions 37 (T to V) and 427 (S to T)	Substitutions in positions 37 (T to V) and 427 (S to T)	Substitutions in positions 37 (T to V) and 427 (S to T).
Type IVB pilus formation outer-membrane protein, R64 PilN family	Absent	Present	Present	Absent
Type VI polysaccharide biosynthesis protein VipB/TviC	Present	Present	Present	Present

Type VI secretion ATPase, ClpV2	Absent	Absent	Absent	T4 phage polysheath and ClpV substrate protein
Type VI secretion protein TssL (Baseplate proteins homologous to T4 phage)	Absent	Baseplate proteins	Absent	Absent
Type VI secretion protein VasK (Transposon-mediated virulence protein)	Present	Absent	Absent	Absent
Type VI secretion system baseplate subunit TssK	Absent	Absent	Absent	Baseplate proteins
Type VI secretion system protein ImpK (Mechanistic contact for inhibition, as well as enhancement)	Substitution of the amino acid from R to N at the 95th position	R-to-N substitution occurred at the 92nd position	At 63, A to G; at 64, N to M; at 67, V to M; at 68, E to D; at 70, V to M; and at 95, R to N	R-to-N substitution at the 95th position
Type VI secretion system protein ImpM	Present	Absent	Present	Present
VipA (TssB) and VipB proteins that form a tubular polymer	Present	Present	Present	Present
Imp family proteins	Present	Present	Present	Present

## Data Availability

Proteome data of all *Pseudomonas fluorescens* strains are deposited in the ProteomeXchange Consortium via the PRIDE [1] partner repository with the dataset identifier PXD019965.

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
