# Peer review of "The Anti-Listeria Activity of Pseudomonas fluorescens Isolated from the Horticultural Environment in New Zealand"

_pathogens, 2023, doi:10.3390/pathogens12020349_

Round 1

Reviewer 1 Report

See attached file.

Author Response

Dear reviewer, Thank you for your time and effort in reviewing our manuscript and we really appreciate the valuable inputs which has immensely improved the quality of our manuscript. We have adapted all the inputs and revised the manuscript and highlighted in the text. We have adapted all your inputs and revised the manuscript, kindly please find the attached comment response and the revised manuscript. Thank you.

Reviewer 2 Report

The authors have submitted a very well organized study and of interest to the wider scientific community.

Comments:

1. I believe that the abstract is too long. Please restructure to make it easier to read.

2. The last part of the introduction sounds more like the aim of choosing the methodology rather than the aim or the hypothesis of the study. Please restructure.

3. section 2 in material and methods needs more detail. The reader needs to be able to reproduce the method. Was the ISO method used?

Author Response

Dear reviewer, Thank you for your valuable time and effort in reviewing our manuscript and we really appreciate the valuable inputs which has immensely improved the quality of our manuscript. We have adapted all the inputs and revised the manuscript and highlighted in the text. Please find the attached response form and the manuscript revised.
